# Does papillary muscle free strain has predictive value in risk stratification of patients with hypertrophic cardiomyopathy?

**Atilla Koyuncu[1], Cennet Yildiz[1‡]\*, Lutfu Ocal[2], Sedat Kalkan[2‡], Alev Kılıçgedik[3], Mustafa Ozan Gürsoy[4‡], Ersan Oflar[1‡], Gökhan Kahveci[5]**

**1** Department of Cardiology, Bakırkoy Dr Sadi Konuk Education and Research Hospital, Istanbul, Turkey, **2** Department of Cardiology, Kosuyolu Education and Research Hospital, Istanbul, Turkey, **3** Department of Cardiology, Basaksehir Cam and Sakura Education and Research City Hospital, Istanbul, Turkey, **4** Department of Cardiology, Izmir Ataturk Education and Research Hospital, Izmir, Turkey, **5** Department of Cardiology, Istinye University, Liv Hospital, Istanbul, Turkey

☉ These authors contributed equally to this work.
‡ CY, SK, MOG and EO also contributed equally to this work.
\* cennet_yildiz@live.com

## Abstract

### Background

Papillary muscle free strain has not been evaluated previously in hypertrophic cardiomyopathy (HCMP) patients. Our aim was to evaluate free papillary muscle free strain in HCMP patients and to find whether it has a value for prediction of sudden cardiac death (SCD) risk score.

### Methods

Transthoracic echocardiography with tissue Doppler imaging, 2-D speckle tracking imaging (STI) of 55 HCMP patients and 45 controls were performed. HCMP patients were further divided into two groups according to their SCD risk score. Patients with risk score of less than 6 points constituted low/intermediate risk group, whereas patients with risk score of greater or equal to 6 points constituted high risk group.

### Results

Interventricular septum, posterior wall, and left ventricular mass index were significantly higher, whereas mitral E/A ratio was significantly lower in HCMP patients compared to controls. Longitudinal apical 4C, 2C, 3C, global longitudinal LV strain, anterolateral papillary muscle (ALPM), posteromedial papillary muscle (PMPM) free strain were significantly reduced in HCMP group compared to control group. Global longitudinal strain and ALPM free strain were significantly lower in patients with high SCD risk score (-14.6 (-17.4 - -13.1) vs -11.6 (-13.2 - -10.1), p = 0.001 and -17.1 (-20.3 - -14.0) vs -9.2 (-12.6 - -7.5), p<0.001, respectively. Global longitudinal strain and ALPM free strain were statistically significantly correlated with SCD risk score (r = 0.480, p<0.001 and r = 0.462, p<0.001, respectively). Global longitudinal strain value of -12.60% had a sensitivity of 73.3% and specificity of

**Data availability statement:** The Data Availability Statement is under discussion and will be provided in a forthcoming update to this article.

**Funding:** The authors received no specific funding for this work.

**Competing interests:** The authors have declared that no competing interests exist.

82.5% for predicting high SCD risk score (AUC: 0.787, 95% CI: 00.643–0.930, p = 0.001). ALPM free strain value of -12.95% had 66.7% sensitivity and 77.5% specificity for predicting high SCD risk score (AUC: 0.766, 95% CI: 0.626–0.905, p = 0.003).

## Conclusion

Papillary muscle free strain was reduced in HCMP patients. It might be used in risk stratification of these patients.

## Introduction

Hypertrophic cardiomyopathy (HCMP) still remains as the most common genetically transmitted heart disease with an estimated prevalence of 1 in 500 persons [1]. With the significant progress that has been made regarding the disease's clinical and genetic perspectives and improvements in diagnostic performance, it has been recognized that the true prevalence might be higher than previous reports [2]. The disease is characterized by myocyte hypertrophy, disarray and interstitial fibrosis that could lead to symptoms including dyspnea, chest pain and palpitation. Generally the prognosis of HCMP is good and majority of the patients have normal life expectancy [3]. However, risk of complications such as sudden cardiac death (SCD), heart failure or atrial fibrillation, are increased in some patients.

Echocardiography has pivotal role in diagnosis as well as management, risk stratification and family screening of HCMP. In the last few decades, technological evolution has enabled us more accurate assessment of cardiac functions with the use of tissue Doppler imaging (TDI), speckle-tracking echocardiography and three-dimensional echocardiography. Several studies have shown that strain parameters were reduced especially in myocardial segments where left ventricular (LV) thickness exceeded 15 mm. HCMP patients with worse strain values might be under increased risk of ventricular arrhythmias, and strain measurement could be used in risk stratification in these patients [4, 5]. Histological and physiological abnormalities in myocardial tissue may serve as a substrate for arrhythmia induction.

Although left ventricular hypertrophy is the predominant phenotypic expression of HCMP, several morphological abnormalities of papillary muscles have been reported between 4 to 13% of HCM patients [6]. Cardiac MRI studies showed that the severity of symptoms, cardiac dysfunction and arrhythmias increase as the abnormalities of papillary muscles increase [7, 8]. Hence, evaluation of papillary muscle anatomy and function has a substantial importance. In the present study, we aimed to evaluate papillary muscle free strain values in HCMP patients and investigate whether it had a predictive value in risk stratification of the disease.

## Material and methods

This was a case-controlled, retrospective study which enrolled 55 HCMP patients and 45 controls. Echocardiographic recordings, clinical, demographical characteristics of the patients were obtained from hospital data base system. Participants with systemic diseases, ischemic heart disease, primary valvular disease, malignancy, thyroid abnormalities, hepatic and/or renal failure, Fabry disease, amyloidosis, Noonan's syndrome, left ventricular outflow gradient (LVOT) less than 30 mmHg at rest and greater or equal to 30 mmHg with Valsalva maneuver were not included in the study. Transthoracic echocardiography with TDI, 2-D speckle tracking imaging (STI) of each patient was evaluated. Ethical committee Bakirkoy Dr. Sadi Konuk Training and Research Hospital approved the study and it was complied in accordance with

the Declaration of Helsinki. All patients' written informed consent were acquired before study enrollment.

Diagnosis of HCMP was done in the presence of LV maximum wall thickness greater than or equal to 15 mm without underlying secondary causes. SCD risk stratification of the patients were calculated according to the current guideline which includes seven variables; age, family history of SCD, maximal LV wall thickness, LA diameter, LVOT gradient, presence of non-sustained ventricular tachycardia (NSVT) and unexplained syncope [9]. Patients were further divided into two groups according to their SCD risk score; patients with risk score of less than 6 points constituted low/intermediate risk group, whereas patients with risk score of greater or equal to 6 points constituted high risk group.

Echocardiographic examinations were performed by iE33 and Q-lab version 8.1 (CMQ, Philips inc). Echocardiography was performed by a cardiologist who had experience in advanced echocardiography and trained for the requirements of the study. All echocardiographic parameters were measured according to current guidelines [10]. Conventional cardiac structural and functional assessment was done by 2-D echocardiography. Interventricular, posterior wall thickness (IVS, PW), left ventricular end-diastolic diameter (LVEDD), left ventricular end-systolic diameter (LVESD), left ventricular mass index (LVMI) were measured. Left ventricular ejection fraction (LVEF) was measured by using modified Simpson method. Early (E) and late (A) mitral diastolic inflow velocities and deceleration time (DT) were obtained by pulsed wave Doppler sample volume which was placed at the tips of mitral and tricuspid valves. Maximal left ventricular outflow tract (LVOT) gradient was measured at rest by using continuous wave Doppler in the apical five-chamber view. Patients who had resting LV outflow gradient less than 30 mmHg underwent Valsalva maneuver in order to elicit latent obstruction. None of the patients in non-obstructive HCMP group had stress pressure gradients more than 30 mmHg. Mitral annular lateral and septal velocities and lateral annular tricuspid velocities were taken by TDI.

Besides from 2-D echocardiographic parameters, 2D-STI was used in order to measure longitudinal systolic strain from apical four-chamber (4-C), two-chamber (2-C) and three-chamber (3-C) views with an increased frame rate of fifty to seventy frames per second. Three to four cardiac cycles from acceptable images were digitally stored for offline analysis in Q-lab software package. Two basal and one apical anchor points were identified manually after which the program traced the endocardial border automatically. In case of inappropriate endocardial tracking, endocardial surfaces were manually corrected by the operator. Apical 4-C, 3-C and 2-C longitudinal strain analyses of each patient were performed. Global longitudinal strain was the average strain value measured by STI with more negative values indicating to higher contractility. Longitudinal myocardial strain of anterolateral and posteromedial papillary muscles (ALPM, PMPM) were obtained by free strain method which evaluates strain values within a myocardial region. In order to calculate papillary muscle free strain, base and tip of papillary muscle was selected manually. Fig 1 depicts the measurement of papillary muscle free strain. All echocardiographic examinations were performed by the same cardiologist who was experienced in echocardiographic imaging. In order to evaluate intra-observer reliability, recordings of 20 patients were re-evaluated 10 days later.

## Statistical analysis

Data was expressed as mean±SD or median (IQR). Two group comparisons were done by independent sample t test or Mann-Whitney U test. ROC curve analysis was conducted in order to find ALPM free strain and global longitudinal strain values for prediction of high

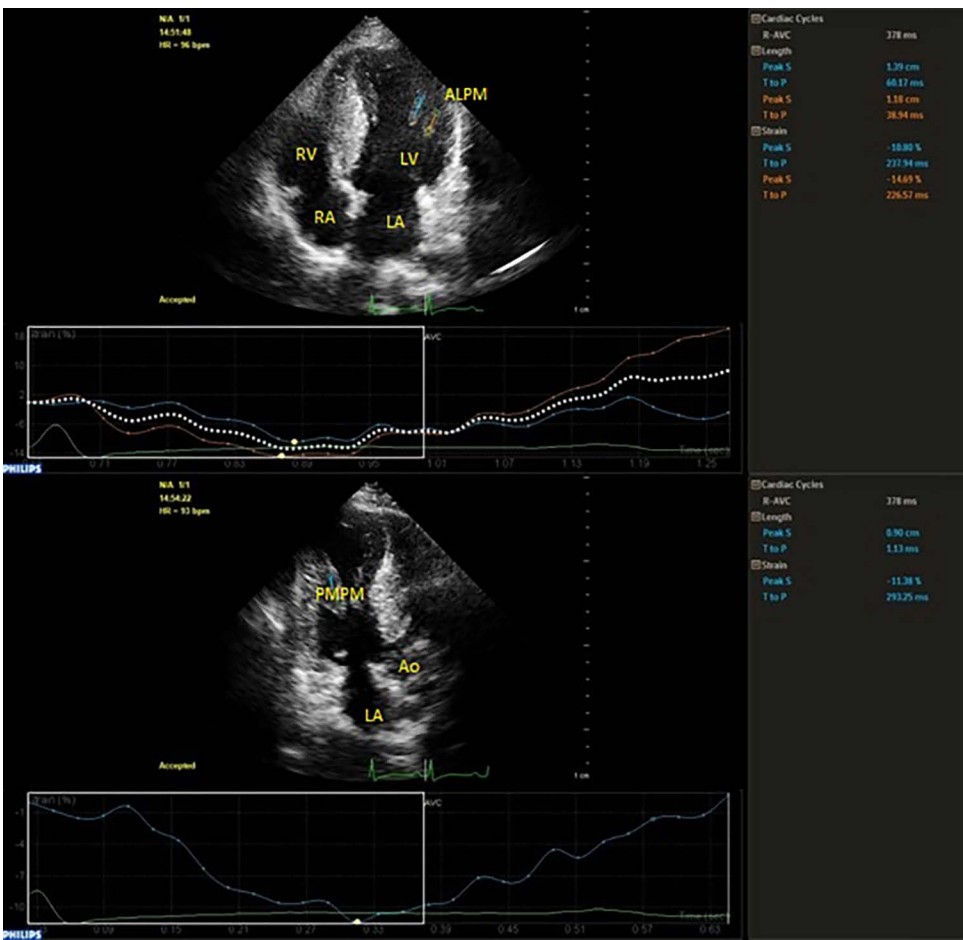

**Fig 1. Measurement of ALPM and PMPM free strain.**

SCD risk score. Correlation of SCD risk score with global longitudinal strain and ALPM free strain were made by use of Spearman correlation analysis. ROC curve analysis was used in order to find the cut-off value of global longitudinal strain and papillary free strain for prediction of high SCD risk score. Independent predictors of high SCD risk score were found by conducting multivariate backward logistic regression analysis. Intra-class correlation coefficient (ICC) model was used for the assessment of difference between two measurements.

## Results and discussion

The median age of the HCMP and the control group were 53.0 (41.0–66.0) years and 54.0 (52.0–56.0) years, respectively. We did not find any differences between two groups with respect to age, gender, body mass index, body surface area of the patients. IVS, PW, LV mass index were significantly higher, whereas mitral E/A ratio was significantly lower in HCMP patients compared to controls. Longitudinal apical 4C, 2C, 3C, global longitudinal LV strain, ALPM, PMPM free strain were significantly reduced in HCMP group compared to control group. Clinical and echocardiographic features of the two groups are given in Table 1.

Fifteen (27.2%) patients with HCMP had history of unexplained syncope; 16 (29.1%) of them had family history of SCD and 9 (16.4%) of them had ICD implantation. Twenty-six (47.27%) patients had LVOT obstruction with a mean maximal gradient of 75.96±30.68

**Table 1. Clinical and echocardiographic characteristics of the two groups.**

| | HCMP group (n = 55) | Control group (n = 45) | p |
|---|---|---|---|
| Gender (n,%) | | | 0.343 |
| Male | 30 (54.5) | 29 (64.4) | |
| Female | 25 (45.5) | 16 (35.6) | |
| Age (years) | 53.0 (41.0–66.0) | 54.0 (52.0–56.0) | 0.775 |
| Body surface area (m²) | 1.8±0.4 | 1.8±0.3 | 0.277 |
| Body mass index (kg/m²) | 26.6±5.4 | 27.2±3.3 | 0.709 |
| IVS (cm) | 2.3 (2.1–2.9) | 0.9 (0.9–1.0) | **<0.001** |
| PW (cm) | 1.3 (1.2–1.5) | 0.8 (0.7–0.8) | **<0.001** |
| LVMI (gr/m²) | 230.8 (162.6–284.6) | 163.3 (138.4–185.3) | **<0.001** |
| TAPSE (mm) | 21.65±6.76 | 25.01±3.37 | 0.244 |
| Mitral E/A | 0.9 (0.7–1.1) | 1.2 (0.8–1.4) | **0.006** |
| Apical longitudinal 4-C strain | -14.5 (-16.3- —11.7) | -38.0 (-46.0- -31.5) | **<0.001** |
| Apical longitudinal 2-C strain | -14.3±3.6 | -19.4±3.2) | **<0.001** |
| Apical longitudinal 3-C strain | -13.4 (-16.2- -10.8) | -42.0 (-16.2- -35.0) | **<0.001** |
| Global longitudinal strain | -14.1±3.2 | -19.9±2.4 | **<0.001** |
| ALPM free strain | -16.0 (-19.0 - —10.0) | -39.0 (-43.5- -34.5) | **<0.001** |
| PMPM free strain | -14.3 (-18.7- -8.1) | -43.0 (-48.0 - -35.5) | **<0.001** |

IVS: Interventricular septum, PW: Posterior wall, LVMI: Left ventricular mass index, TAPSE: Tricuspid annular plane systolic excursion, ALPM: anterolateral papillary muscle, PMPM: Posteromedial papillary muscle.

mmHg. The type of hypertrophy was asymmetrical septal, concentric and apical in 37 (67.27%), 14 (25.45) and 4 (7.2%) patients, respectively. Fifteen patients (27.2%) had ALPM abnormalities (apical insertion of papillary muscles, bifid papillary muscle), seven (12.7%) patients had PMPM abnormalities and 5 (9%) patients had both ALPM and PMPM abnormalities.

The patients were further compared according to their SCD risk score. Although IVS, PW thickness, maximal wall thickness, left atrial size and LVMI were found to be higher in SCD high risk patients, these values did not reach statistical significance. Patients who had high risk scores for SCD had significantly higher values of left ventricular outflow gradient, higher rates of family history of SCD, non-sustained VT and syncope history. These patients also had lower values of mitral lateral annular S' velocity. We did not find any differences in other clinical, 2-D and TDI variables between two groups of the patients. However, apical 4-C, 2-C, 3-C and global longitudinal strain and ALPM free strain were significantly impaired in high risk patients with HCMP (Table 2).

Global longitudinal strain and ALPM free strain were statistically significantly correlated with SCD risk score (r = 0.480, p<0.001 and r = 0.462, p<0.001, respectively) (Figs 2 and 3, respectively). Moreover, ALPM free strain showed moderate correlation with maximal LV wall thickness (r = 0.406, p = 0.002). Global longitudinal strain value of -12.60% had a sensitivity of 73.3% and specificity of 82.5% for predicting high SCD risk score (AUC: 0.787, 95% CI: 0.643–0.930, p = 0.001). ALPM free strain value of -12.95% had 66.7% sensitivity and 77.5% specificity for predicting high SCD risk score (AUC: 0.766, 95% CI: 0.626–0.905, p = 0.003) (Fig 4). Results of backward logistic regression analysis showed that only age, global longitudinal strain and ALPM free strain were the predictors of high SCD risk score (Table 3).

ALPM and PMPS values of the patients were analyzed according to the presence of papillary muscle abnormalities. ALPM free strain values were significantly reduced in patients who had morphological abnormalities of papillary muscles compared to patients who had no

**Table 2. Comparison of the patients with low and high SCD risk score.**

| | Low/intermediate risk group for SCD (n = 40) | High risk group for SCD (n = 15) | p |
|---|---|---|---|
| Gender (n, %) | | | 0.265 |
| Male | 20 (50) | 10 (66.7) | |
| Female | 20 (50) | 5 (33.3) | |
| Age (years) | 54.9±16.0 | 45.6±14.6 | 0.056 |
| Body surface area (m²) | 1.8±0.4 | 1.7±0.3 | 0.317 |
| Body mass index (kg/m²) | 27.2.±6 | 25.6±3.3 | 0.212 |
| IVS (cm) | 2.3 (2.1–2.9) | 2.9 (2.1–3.2) | 0.211 |
| PW (cm) | 1.3 (1.1–1.5) | 1.4 (1.3–1.7) | 0.065 |
| IVS/PW | 1.9±0.6 | 1.8±0.5 | 0.533 |
| Max Wall thickness (cm) | 2.3 (2.1–2.9) | 2.9 (2.3–3.2) | 0.225 |
| LA (cm) | 3.9±6.3 | 4.1±7.0 | 0.395 |
| LVOT gradient (mmHg) | 20.5 (11–59.2) | 65 (15–101) | **0.032** |
| Family history of SCD (n, %) | 6 (15) | 10 (66.7) | **<0.001** |
| Non-sustained VT (n, %) | 4 (10) | 9 (60) | **<0.001** |
| Syncope (n, %) | 4 (10) | 11 (73.3) | **<0.001** |
| LVMI (gr/m²) | 215.8 (155.8–260.6) | 249.6 (180.1–404.9) | 0.156 |
| LVEF (%) | 55.5 (52.2–58.4) | 57.2 (55.3–59.5) | 0.195 |
| TAPSE (mm) | 21.89±6.6 | 21.8±5.2 | 0.956 |
| Mitral E/A | 0.9 (0.7–1.1) | 0.9 (0.6–1.1) | 0.985 |
| Mitral Tei Index | 0.6±0.1 | 0.6±0.1 | 0.431 |
| Tricuspid E/A | 1.1 (0.9–1.3) | 1.0 (0.9–1.3) | 0.438 |
| Mitral septal S' | 7.1±1.8 | 6.4±2.3 | 0.197 |
| Mitral septal E' | 5.3 (3.8–7.5) | 5.6 (4.2–7.2) | 0.650 |
| Mitral septal A' | 7.7±2.3 | 7.0±2.8 | 0.367 |
| Mitral lateral S' | 8.4±1.9 | 6.8±2.2 | **0.014** |
| Mitral lateral E' | 7.5 (5.6–9.0) | 7.6 (6.2–9.1) | 0.650 |
| Mitral lateral A' | 8.7±3.2 | 8.4±3.3 | 0.844 |
| Tricuspid S' | 13.0 (11.9–17.4) | 13.1 (10.1–16) | 0.420 |
| Tricuspid E' | 10.7±3.4 | 11.2±4.7 | 0.666 |
| Tricuspid A' | 14.1±5.1 | 15.3±4.8 | 0.486 |
| 3D-LAVI | 42.3 (35.7–51.4) | 41.0 (33.5–56.8) | 0.868 |
| Apical longitudinal 4-C strain | -15.4.9±3.4 | -12.4±3.2 | **0.015** |
| Apical longitudinal 2-C strain | -15.1±3.4 | -12.22±3.2 | **0.008** |
| Apical longitudinal 3-C strain | -14.4±3.7 | -11.1±3.5 | 0.006 |
| Global longitudinal strain | -14.9±2.9 | -11.7±3.0 | **0.001** |
| ALPM free strain | -17.1 (-20.3 - -14.0)) | -9.2 (-12.6 - -7.5) | **<0.001** |
| PMPM free strain | -15.0 (-19.0- —11.8) | -8.6 (-17.8- —7.6) | 0.074 |

IVS: Interventricular septum, PW: Posterior wall, LVMI: Left ventricular mass index, LVEF: Left ventricular ejection fraction, LVOT: Left ventricular outflow tract, TAPSE: Tricuspid annular plane systolic excursion, LAVI: Left atrial volume index, ALPM: anterolateral papillary muscle, PMPM: Posteromedial papillary muscle.

abnormalities in papillary muscles (-13.87±4.93 vs -19.50±8.11, p = 0.022). Although PMPM free strain values were reduced in patients who had PMPM abnormalities, it did not reach statistical significance (-14.68±6.73 vs -10.48±17.87, p = 0.253).

In order to measure intra-observer variability, global longitudinal strain measurement of 15 patients were performed twice with 10 days interval. Intra-observer variability of two measures was found to be as 0.868 (CI 95%: 0.637–0.950).

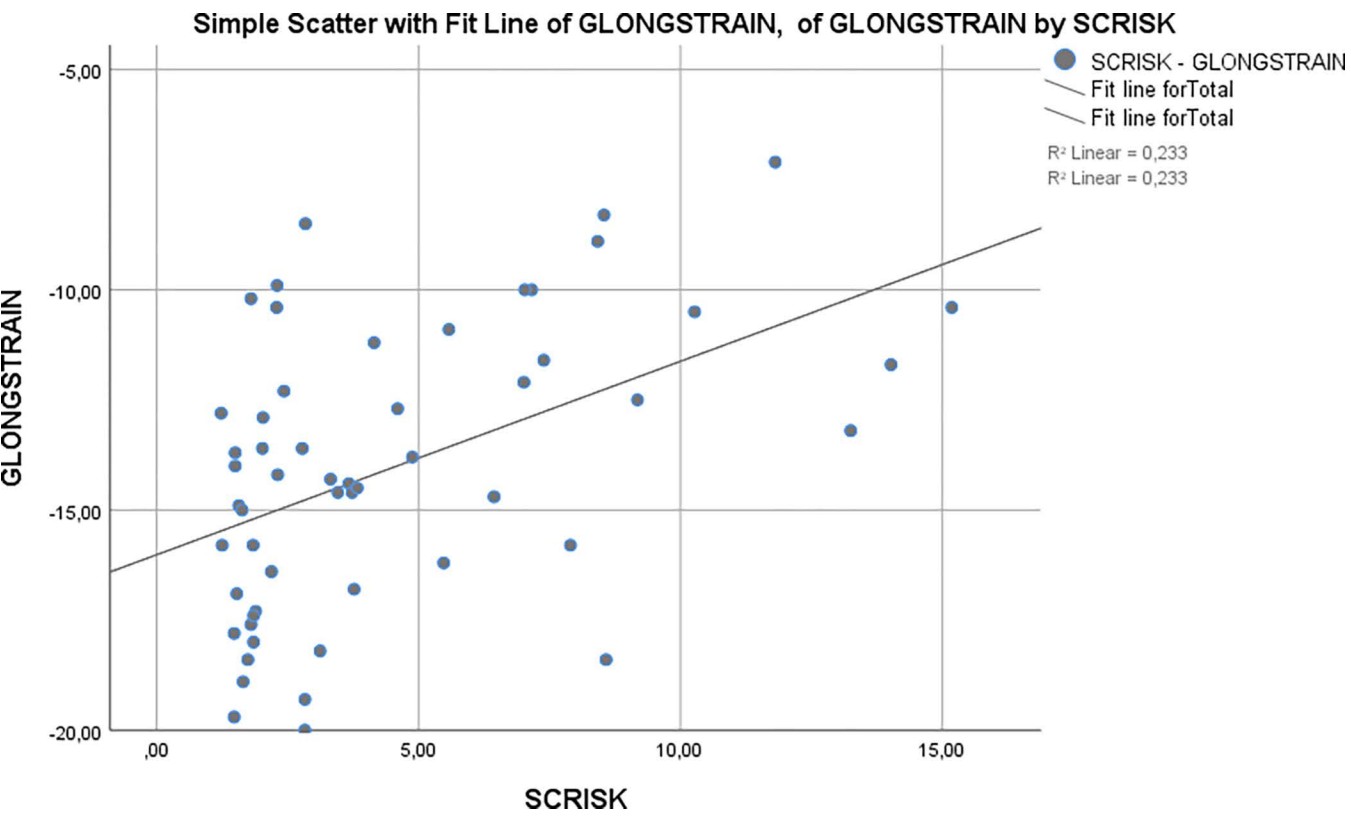

**Fig 2. Correlation of global longitudinal strain with SCD risk score.**

## Discussion

In the present study, we evaluated papillary muscle free strain values in patients with HCMP. The results of our study suggested that papillary muscle free strain values were significantly reduced in HCMP patients compared to controls. Moreover, patients who had higher SCD risk scores had more profound reduction in papillary muscle free strain values indicating greater impairment.

Theoretically, HCMP may involve any location in the heart, but IVS is by far the most commonly affected segment, accounting for 50% of all cases [11]. Similarly, in the present study IVS was the most commonly affected site followed by concentric and apical involvement. Almost one third (30.9%) of patients had papillary muscle abnormalities including papillary muscle hypertrophy, anteroapical displacement of papillary muscles or bifid papillary muscle.

Two-dimensional STI provides information about global or regional myocardial deformation mechanics. It allows us to detect early myocardial functional abnormalities before significant changes in LVEF are apparent [12]. Previous studies have shown a clear association between myocardial fibrotic changes and reduced strain values in HCMP patients [13, 14]. In a study of Yang et al. degree of reduction in strain values was greater in hypertrophied cardiac segments and correlated with the amount of histopathologic abnormalities [15]. Almaas et al. compared strain imaging with magnetic resonance imaging in detection of fibrosis in HCMP patients who underwent septal myectomy. They studied myectomy specimens and histological fibrosis was classified as interstitial, replacement and total. According to their findings, septal, interstitial or replacement fibrosis did not correlate with late gadolinium enhancement on cardiac MRI. However septal longitudinal strain showed moderate correlation with total

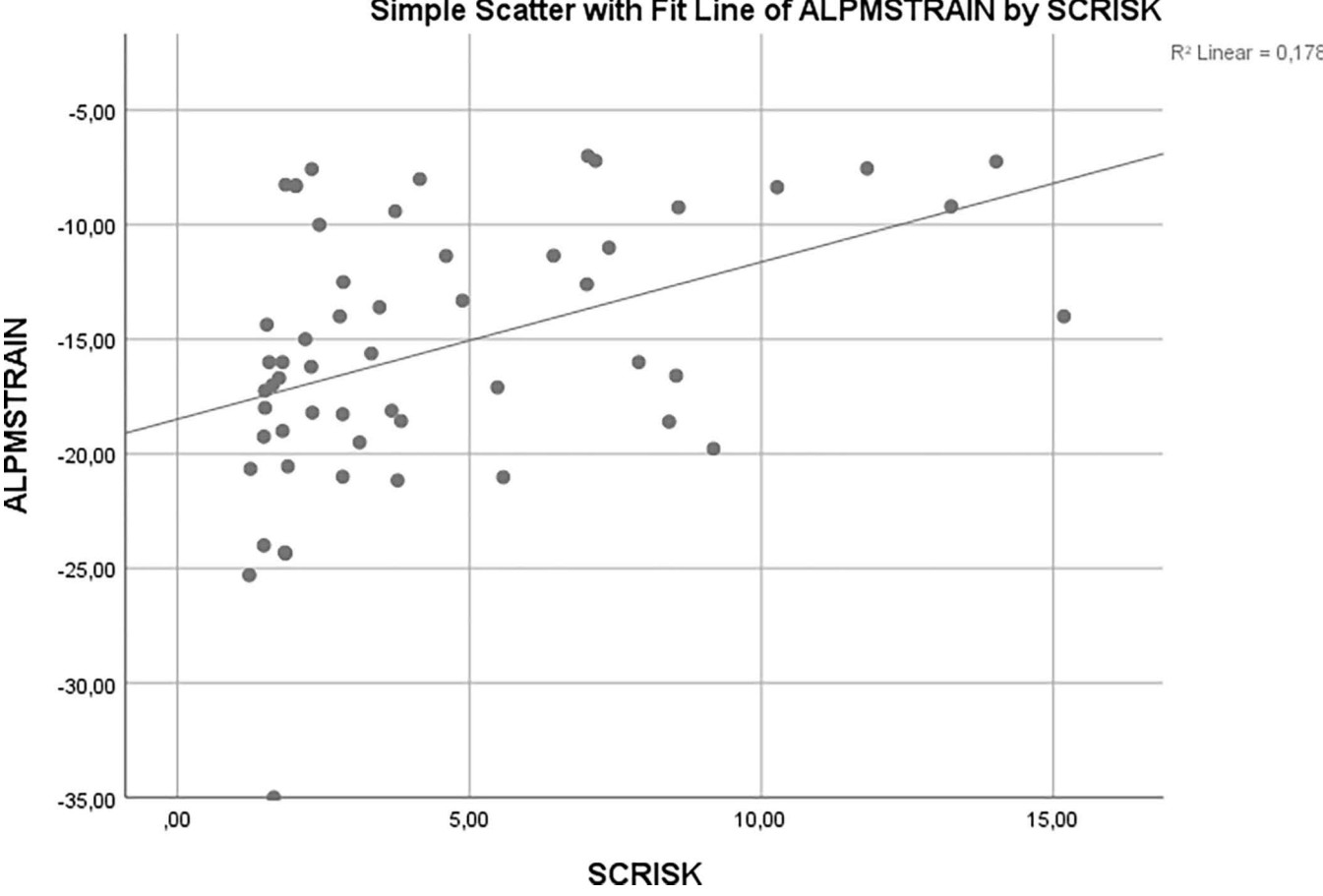

**Fig 3. Correlation of ALPM free strain with SCD risk score.**

and interstitial fibrosis. Moreover, reduced longitudinal septal strain predicted ventricular arrhythmias in this group of subjects. They suggested that myocardial functional assessment by strain might be more robust tool in assessment of fibrosis compared to visual assessment by cardiac MRI [16]. Our study was in concordance with the aforementioned studies in which significant reduction in global longitudinal strain values were found in HCMP patients. Global longitudinal strain had a positive correlation with SCD risk score.

Although the predominant characteristic feature of HCMP is left ventricular hypertrophy, papillary muscle abnormalities have been reported more frequently in the recent years. These abnormalities might be one of the underlying mechanism for LVOT obstruction which has been traditionally attributed to septal hypertrophy and Venturi effect [17–21]. Kwon et al. indicated that association between morphological abnormalities of papillary muscles and LVOT obstruction/systolic anterior motion of mitral valve was independent from septal hypertrophy [22]. Variable associations between papillary muscle fibrotic changes and patients' symptoms and cardiac arrhythmias have been reported in late gadolinium enhancement MRI studies [23]. According to our results, ALPM and PMPM free strain were reduced in HCMP patients. ALPM free strain was significantly reduced in patients who have high risk for SCD. Although PMPM free strain was lower in patients with high SCD risk, it did not reach statistical significance. The patients with ALPM abnormalities had more reduced strain values in the affected papillary muscles compared to patients who had normal anatomy.

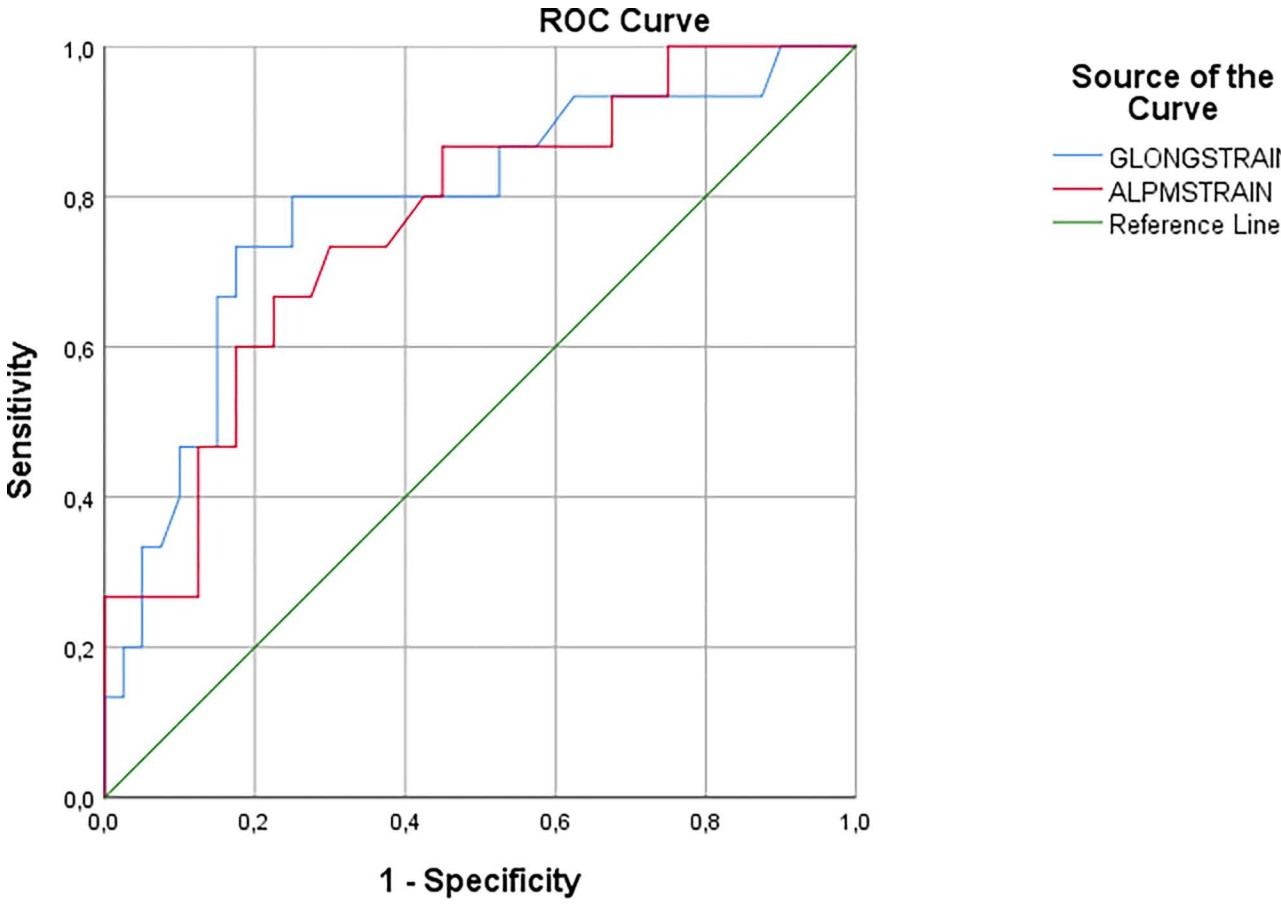

**Fig 4. ROC curve analysis of global longitudinal strain and ALPM free strain for prediction of high SCD risk score.**

The present study indicated that papillary muscle functions were reduced in HCMP patients. Previous studies reported that the frequency of myocardial fibrosis and fiber disarray did not significantly differ between left ventricular sections and their presence was independent of wall and septal thickness [24]. This may be the reason for reduced papillary muscle free strain values in our study. In addition, patients with higher risk scores for SCD had more reduced strain values pointing to more severe involvement by the disease. Additionally, our results showed higher incidence of ALPM abnormalities compared to PMPM which had a predictive value for the SCD risk score. SCD risk stratification in HCM patients is based on family history, patients' symptoms and echocardiographic variables. However, new risk markers for adverse outcomes have been proposed in the past few decades, such as presence of late gadolinium enhancement in cardiac MRI or abnormal strain values in STI studies [25, 26]. It has been shown that presence of late gadolinium enhancement on cardiac MRI was associated with adverse cardiac outcomes including ventricular arrhythmias, sudden cardiac death, heart failure symptoms and reduced left ventricular ejection fraction [27–29]. Late gadolinium enhancement of more than 15% in cardiac MRI studies was related to increased risk of SCD in HCMP patients [28, 30]. According to our results, papillary muscle free strain may give us information about papillary muscle function and help for risk stratification of these patients.

**Table 3. Backward multivariate logistic regression for predictors of SCD.**

|  | Parameter | P | OR | 95% CI |
|---|---|---|---|---|
| STEP 1 | Age | 0.020 | 0.926 | 0.868–0.988 |
|  | Max Wall Thickness | 0.055 | 0.822 | 0.672–1.005 |
|  | LA | 0.196 | 1.101 | 0.951–1.274 |
|  | LVOT gradient | 0.073 | 1.022 | 0.998–1.046 |
|  | Global longitudinal strain | 0.027 | 1.261 | 1.027–1.548 |
|  | ALPM free strain | 0.079 | 1.185 | 0.981–1.432 |
| STEP 2 | Age | 0.030 | 0.934 | 0.879–0.994 |
|  | Max Wall Thickness | 0.077 | 0.851 | 0.928–1.215 |
|  | LVOT gradient | 0.111 | 1.018 | 0.996–1.041 |
|  | Global longitudinal strain | 0.010 | 1.563 | 1.111–2.200 |
|  | ALPM free strain | 0.028 | 1.257 | 1.025–1.541 |
| STEP 3 | Age | 0.019 | 0.927 | 0.871–0.988 |
|  | Max Wall Thickness | 0.126 | 0.884 | 0.755–1.035 |
|  | Global longitudinal strain | 0.008 | 1.526 | 1.114–2.090 |
|  | ALPM free strain | 0.018 | 1.279 | 1.043–1.569 |
| STEP 4 | Age | 0.040 | 0.946 | 0.898–0.997 |
|  | Global longitudinal strain | 0.017 | 1.409 | 1.063–1.869 |
|  | ALPM free strain | 0.041 | 1.208 | 1.008–1.448 |

LVOT: Left ventricular outflow tract, ALPM: anterolateral papillary muscle.

## Limitations

It was a single-center study with a relatively low sample size. Since the long term follow-up of the patients was not available, we did not know the prognostic value of papillary muscle free strain. Comparison of echocardiographic variables of the patients with cardiac MRI findings were not made.

## Conclusions

Our study confirmed papillary muscle involvement in HCMP. Functional abnormalities of papillary muscles might be used in risk stratification of patients. Further studies are needed in order to evaluate the prognostic significance of papillary muscle free strain in HCMP patients.

## Author contributions

**Conceptualization:** Atilla Koyuncu, Lutfu Ocal, Sedat Kalkan, Alev Kılıçgedik, Mustafa Ozan Gürsoy, Ersan Oflar, Gökhan Kahveci.

**Data curation:** Atilla Koyuncu, Lutfu Ocal, Alev Kılıçgedik, Mustafa Ozan Gürsoy, Ersan Oflar, Gökhan Kahveci.

**Formal analysis:** Atilla Koyuncu, Cennet Yildiz, Mustafa Ozan Gürsoy.

**Funding acquisition:** Lutfu Ocal, Alev Kılıçgedik.

**Investigation:** Atilla Koyuncu, Cennet Yildiz, Lutfu Ocal, Sedat Kalkan, Alev Kılıçgedik, Ersan Oflar, Gökhan Kahveci.

**Methodology:** Atilla Koyuncu, Cennet Yildiz, Lutfu Ocal, Sedat Kalkan, Alev Kılıçgedik, Ersan Oflar.

**Project administration:** Sedat Kalkan.

**Resources:** Gökhan Kahveci.

**Software:** Mustafa Ozan Gürsoy.

**Supervision:** Atilla Koyuncu, Cennet Yildiz, Lutfu Ocal, Sedat Kalkan, Mustafa Ozan Gürsoy, Ersan Oflar.

**Validation:** Sedat Kalkan, Ersan Oflar.

**Writing – original draft:** Cennet Yildiz, Ersan Oflar.

**Writing – review & editing:** Atilla Koyuncu, Alev Kılıçgedik, Mustafa Ozan Gürsoy, Ersan Oflar, Gökhan Kahveci.

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
