## [Decision Letter · Decision Letter 0]

19 Sep 2022

PONE-D-22-24350

Does papillary muscle free strain has predictive value in risk stratification of patients with hypertrophic cardiomyopathy?

PLOS ONE

Dear Dr. YILDIZ,

Thank you for submitting your manuscript to PLOS ONE. The MS was reviewed by two referees. One requested major revision, the other wanted the MS to be rejected. My impression is also major revision, but this revision must be done very thoroughly and address sufficiently all the points the reviewers raised. In addition: My concern is the correlation strain vs. sudden cardiac death. Though mathematically correct (I assume) the data points show rather uncorrelated data in the low and mid risk group. Only the high risk group shows higher strains. However, these values are not much higher those in the upper range of the low and mid risk group. Or to put it the other way round: if you find a high strain, you do not know much about the risk of the patient. Only when the strain is appropriately low, the risk of sudden cardiac death is low. So strain has a more negative prognostic value.  Please comment on this. 

We look forward to receiving your revised manuscript.

Kind regards,

Wolfgang Rudolf Bauer, M.D., Ph.D.

Academic Editor

PLOS ONE

Journal Requirements:

a) Did participants provide their written or verbal informed consent to participate in this study?

4. In your Data Availability statement, you have not specified where the minimal data set underlying the results described in your manuscript can be found. PLOS defines a study's minimal data set as the underlying data used to reach the conclusions drawn in the manuscript and any additional data required to replicate the reported study findings in their entirety. All PLOS journals require that the minimal data set be made fully available. For more information about our data policy, please see http://journals.plos.org/plosone/s/data-availability .

"Upon re-submitting your revised manuscript, please upload your study’s minimal underlying data set as either Supporting Information files or to a stable, public repository and include the relevant URLs, DOIs, or accession numbers within your revised cover letter. For a list of acceptable repositories, please see http://journals.plos.org/plosone/s/data-availability#loc-recommended-repositories . Any potentially identifying patient information must be fully anonymized.

Important: If there are ethical or legal restrictions to sharing your data publicly, please explain these restrictions in detail. Please see our guidelines for more information on what we consider unacceptable restrictions to publicly sharing data: http://journals.plos.org/plosone/s/data-availability#loc-unacceptable-data-access-restrictions . Note that it is not acceptable for the authors to be the sole named individuals responsible for ensuring data access.

Reviewers' comments:

Reviewer's Responses to Questions

**Comments to the Author**

1. Is the manuscript technically sound, and do the data support the conclusions?

Reviewer #1: Partly

Reviewer #2: No

2. Has the statistical analysis been performed appropriately and rigorously?

Reviewer #1: No

Reviewer #2: No

3. Have the authors made all data underlying the findings in their manuscript fully available?

Reviewer #1: No

Reviewer #2: Yes

4. Is the manuscript presented in an intelligible fashion and written in standard English?

Reviewer #1: Yes

Reviewer #2: Yes

5. Review Comments to the Author

Reviewer #1: In the present manuscript the authors try to answer the question whether papillary muscle free strain predicts the value in risk stratification of patients with hypertrophic cardiomyopathy? Beside the fact that it would be more important to analyze the potential influence on observed risk there some points that should be discussed.

They classified patients using the ESC risk calculator which tries to calculate the risk of sudden cardiac death (SCD) within the next 5 years. The model was developed in order to help clinicians in decision making of indication for ICD implantation in primary prevention.

The authors didn’t use the reported risk thresholds properly. High risk patients according ESC risk calculation model are defined as patients with a 5 years risk of SCD >6% - and NOT >4%. The model defines a risk >4 and <6% as patients of medium risk. Therefore, it is important to analyze the data using thresholds and definitions given in the guidelines.

Furthermore, figure 2 should use another grading with steps of 2% calculated risk for SCD. The reported correlation is weak.

It would be of clinical interest to give the values of the single risk factors which are used in the risk stratification model.

Can the authors give information about patient follow-up including real risk of SCD.

It does not make sense to give the percentages with 2 decimal places if only 55 patients were examined.

Did they measure LV gradients at rest or provocation?

The chapter discussion includes some incorrect information like “Moreover almost one-quarter of HCMP patients had remaining high gradient after septal myectomy” or “However, in the past few decades histological abnormalities such as late gadolinium enhancement in cardiac MR studies or abnormalities in strain values have emerged as a new risk markers for adverse outcomes”

Reviewer #2: Does papillary muscle free strain has predictive value in risk stratification of patients with hypertrophic cardiomyopathy?

Koyunzu et al. analyzed the transthoracic echocardiograms of 55 HCMP patients and 45 controls using 2D speckle-tracking imaging. The 55 HCMP patients were further subdivided according to their sudden cardiac death risk score (=4 points).

The authors found higher wall thickness and worse diastolic dysfunction in HCMP patients when compared to controls. They further found global longitudinal strain as well as papillary muscle strain values significantly different in HCMP patients. Global longitudinal strain and ALPM free strain were predictive of a high risk of SCD as calculated by the risk score with a low sensitivity but moderate to high specificity.

The manuscript deals with an iteresting topic. Nevertheless, there are several points to address:

Abstract:

• In the results part, a „-„ is missing before „11.97“.

• „ […]-15.88±12.73 vs -11.71±10.40, p=0.163, respectively“: either the p-value or the statement about significant group differences is not correct. Please revise.

• „(-16.70 (-19.37- -11.93) vs 11.97 (-18.57 - -7.90); p= 0.048 and -15.88±12.73 vs -11.71±10.40, p=0.163, respectively)“: it is unclear, which strain these values refer to? The global longitudinal strain or the two papillary muscle strains. Further, it is a bit confusing to present the one strain value as median and the other as mean. I would suggest to present both as median, if the mean cannot be presented due to skewed distribution.

Manuscript:

• Page and line numbering would have been very helpful…

Material and Methods:

• How did you identifiy the patients? Did you include consecutive patients presenting to your hospital or did you search the hospital clinical information system? How did you acquire informed consent?

• How did you identify the controls?

• „SCD risk stratification of the patients were calculated according to the current guideline which includes five variables“ … there are 7 variables listed…

• Were the echocardiograms performed according to a certain standard protocol? Who performed the echos?

Results:

• „The mean age of the HOCM and the control group were […]“ you might have meant HCMP instead of HOCM?

• „Intra-observer variability of two measures was found to be as 0.868 (CI 95%: 0.637-0.950).“ Please indicate, which parameter was analyzed. Usually, intra-observer variability varies for each parameter.

• Maximal LV wall thickness, LA diameter, and left ventricular outflow tract (LVOT) gradient, which all entered the risk score, have a strong relationship with LV strain values. A sensitivity analysis without these variables might be interesting to show an independent association of strain values and SCD risk.

• Did you perform a follow-up to longitudinally assess the true risk of SCD/ICD shock?

• Please provide the diagnostic accuracy (sens/spec) of the papillary muscle strain when adjusted for other variables of known prongostic value (multi-marker model including GLPS, LVEF, E/e´, NT-proBNP, troponin, …). Is there an incremental value of these two strain values over known parameters indicating higher risk?

• Was there an association of maximal LV wall thickness and papillary muscle strain?

• Was an MRI available at least in some patients? Was there an association of LGE and papillary muscle strain?

In total, the incremental value of measuring papillary muscle strain, which is a sophisticated and time consuming method, does not become overt, so far.

6. PLOS authors have the option to publish the peer review history of their article (what does this mean? ). If published, this will include your full peer review and any attached files.

**Do you want your identity to be public for this peer review?** For information about this choice, including consent withdrawal, please see our Privacy Policy .

Reviewer #1: No

Reviewer #2: No

---

## [Author Response · Author response to Decision Letter 0]

7 Nov 2022

PONE-D-22-24350

Does papillary muscle free strain has predictive value in risk stratification of patients with hypertrophic cardiomyopathy?

PLOS ONE

Dear Dr. YILDIZ,

Thank you for submitting your manuscript to PLOS ONE. The MS was reviewed by two referees. One requested major revision, the other wanted the MS to be rejected. My impression is also major revision, but this revision must be done very thoroughly and address sufficiently all the points the reviewers raised.

In addition: My concern is the correlation strain vs. sudden cardiac death. Though mathematically correct (I assume) the data points show rather uncorrelated data in the low and mid risk group. Only the high risk group shows higher strains. However, these values are not much higher those in the upper range of the low and mid risk group. Or to put it the other way round: if you find a high strain, you do not know much about the risk of the patient. Only when the strain is appropriately low, the risk of sudden cardiac death is low. So strain has a more negative prognostic value. Please comment on this.

We look forward to receiving your revised manuscript.

Kind regards,

Wolfgang Rudolf Bauer, M.D., Ph.D.

Academic Editor

PLOS ONE

Journal Requirements:

• Manuscript has been revised according to PLOS ONE’s style requirements.

a) Did participants provide their written or verbal informed consent to participate in this study?

• All the participants provided their written informed consent to participate in this study.

• Financial disclosure has been stated in the cover letter.

4. In your Data Availability statement, you have not specified where the minimal data set underlying the results described in your manuscript can be found. PLOS defines a study's minimal data set as the underlying data used to reach the conclusions drawn in the manuscript and any additional data required to replicate the reported study findings in their entirety. All PLOS journals require that the minimal data set be made fully available. For more information about our data policy, please see http://journals.plos.org/plosone/s/data-availability .

"Upon re-submitting your revised manuscript, please upload your study’s minimal underlying data set as either Supporting Information files or to a stable, public repository and include the relevant URLs, DOIs, or accession numbers within your revised cover letter. For a list of acceptable repositories, please see http://journals.plos.org/plosone/s/data-availability#loc-recommended-repositories . Any potentially identifying patient information must be fully anonymized.

Important: If there are ethical or legal restrictions to sharing your data publicly, please explain these restrictions in detail. Please see our guidelines for more information on what we consider unacceptable restrictions to publicly sharing data: http://journals.plos.org/plosone/s/data-availability#loc-unacceptable-data-access-restrictions . Note that it is not acceptable for the authors to be the sole named individuals responsible for ensuring data access.

• Minimal data set of the study have been available as Supporting Information files.

• Full name of the ethic committee has been provided in the Methods section. All patients gave written informed consent before study enrollment.

Reviewers' comments:

Reviewer's Responses to Questions

Comments to the Author

1. Is the manuscript technically sound, and do the data support the conclusions?

Reviewer #1: Partly

Reviewer #2: No

2. Has the statistical analysis been performed appropriately and rigorously?

Reviewer #1: No

Reviewer #2: No

3. Have the authors made all data underlying the findings in their manuscript fully available?

Reviewer #1: No

Reviewer #2: Yes

4. Is the manuscript presented in an intelligible fashion and written in standard English?

Reviewer #1: Yes

Reviewer #2: Yes

5. Review Comments to the Author

Reviewer #1: In the present manuscript the authors try to answer the question whether papillary muscle free strain predicts the value in risk stratification of patients with hypertrophic cardiomyopathy? Beside the fact that it would be more important to analyze the potential influence on observed risk there some points that should be discussed.

They classified patients using the ESC risk calculator which tries to calculate the risk of sudden cardiac death (SCD) within the next 5 years. The model was developed in order to help clinicians in decision making of indication for ICD implantation in primary prevention.

The authors didn’t use the reported risk thresholds properly. High risk patients according ESC risk calculation model are defined as patients with a 5 years risk of SCD >6% - and NOT >4%. The model defines a risk >4 and <6% as patients of medium risk. Therefore, it is important to analyze the data using thresholds and definitions given in the guidelines. Furthermore, figure 2 should use another grading with steps of 2% calculated risk for SCD. The reported correlation is weak.

• We thank the reviewer. We recalculated the risk of the patients according to ESC risk calculation method (which includes seven variables) and divided into two groups according to their risks (low-intermediate risk group and high risk group). Our analysis showed that global longitudinal strain and anterior papillary muscle free strain were significantly reduced in patients with high risk scores. Correlation of sudden cardiac risk score with global longitudinal strain and anterolateral papillary muscle free strain have been depicted in Figure 2 and Figure 3, respectively. We found moderate correlation between sudden cardiac risk score and global longitudinal strain and anterolateral papillary muscle free strain.

It would be of clinical interest to give the values of the single risk factors which are used in the risk stratification model.

• We analyzed the single risk factors which were used in risk stratification model. According to our analysis patients who had high risk for SCD had higher LVOT gradient, higher rate of family history of SCD, non-sustained VT and syncope history.

Can the authors give information about patient follow-up including real risk of SCD.

• Unfortunately long-term follow-up of the patients was not done in the present study.

It does not make sense to give the percentages with 2 decimal places if only 55 patients were examined.

• We thank the reviewer. The percentages was given with one decimal places.

Did they measure LV gradients at rest or provocation?

• We thank the reviewer. LV gradients were measured at rest. It has been stated in the Methods section.

The chapter discussion includes some incorrect information like “Moreover almost one-quarter of HCMP patients had remaining high gradient after septal myectomy” or “However, in the past few decades histological abnormalities such as late gadolinium enhancement in cardiac MR studies or abnormalities in strain values have emerged as a new risk markers for adverse outcomes”

• We thank the reviewer. Incorrect information have been corrected. The sentence “Moreover almost one-quarter of HCMP patients had remaining high gradient after septal myectomy” has been removed from the discussion section.

Reviewer #2: Does papillary muscle free strain has predictive value in risk stratification of patients with hypertrophic cardiomyopathy?

Koyunzu et al. analyzed the transthoracic echocardiograms of 55 HCMP patients and 45 controls using 2D speckle-tracking imaging. The 55 HCMP patients were further subdivided according to their sudden cardiac death risk score (=4 points).

The authors found higher wall thickness and worse diastolic dysfunction in HCMP patients when compared to controls. They further found global longitudinal strain as well as papillary muscle strain values significantly different in HCMP patients. Global longitudinal strain and ALPM free strain were predictive of a high risk of SCD as calculated by the risk score with a low sensitivity but moderate to high specificity.

The manuscript deals with an iteresting topic. Nevertheless, there are several points to address:

Abstract:

• In the results part, a „-„ is missing before „11.97“.

• „ […]-15.88±12.73 vs -11.71±10.40, p=0.163, respectively“: either the p-value or the statement about significant group differences is not correct. Please revise.

• „(-16.70 (-19.37- -11.93) vs 11.97 (-18.57 - -7.90); p= 0.048 and -15.88±12.73 vs -11.71±10.40, p=0.163, respectively)“: it is unclear, which strain these values refer to? The global longitudinal strain or the two papillary muscle strains. Further, it is a bit confusing to present the one strain value as median and the other as mean. I would suggest to present both as median, if the mean cannot be presented due to skewed distribution.

• We thank the reviewer. All the numbers, p values have been revised and corrected. All the numbers in the abstract section were given as median and IQR.

Manuscript:

Page and line numbering would have been very helpful…

• We thank the reviewer. Page and line numbers have been added.

Material and Methods:

How did you identifiy the patients? Did you include consecutive patients presenting to your hospital or did you search the hospital clinical information system? How did you acquire informed consent?

• We thank the reviewer. In our hospital, echocardiographic examinations of the patients who had specific diseases were recorded and stored. Before echocardiographic examination, written informed consent of each patient was obtained. In the present study, we included consecutive patients who applied to our echocardiography department with the diagnosis of hypertrophic cardiomyopathy. Patients with poor image quality were excluded from the study.

How did you identify the controls?

• We thank the reviewer. After selection of HCMP patients, age and sex matched healthy controls were included in the study.

„SCD risk stratification of the patients were calculated according to the current guideline which includes five variables“ … there are 7 variables listed…

• We thank the reviewer. SCD risk stratification was done according to ESC guideline which includes seven variables. It has been corrected. SCD risk of each patient has been recalculated.

Were the echocardiograms performed according to a certain standard protocol? Who performed the echos?

• We thank the reviewer. Echocardiography was performed by a cardiologist who had experience in advanced echocardiography and trained for the requirements of the study. All of the echocardiographic examinations were done according to current guidelines.

Results:

„The mean age of the HOCM and the control group were […]“ you might have meant HCMP instead of HOCM?

• We thank the reviewer. It has been corrected.

„Intra-observer variability of two measures was found to be as 0.868 (CI 95%: 0.637-0.950).“ Please indicate, which parameter was analyzed. Usually, intra-observer variability varies for each parameter.

• We thank the reviewer. For evaluation of intra-observer variability, global longitudinal strain measurements were used. It has been stated in results section.

• Maximal LV wall thickness, LA diameter, and left ventricular outflow tract (LVOT) gradient, which all entered the risk score, have a strong relationship with LV strain values. A sensitivity analysis without these variables might be interesting to show an independent association of strain values and SCD risk.

• We thank the reviewer. In order to found the independent predictors of high SCD risk scores backward logistic regression analysis (with inclusion of parameters: Maximal LV wall thickness, LA diameter, and left ventricular outflow tract (LVOT) gradient) was conducted. The results showed that only age global longitudinal strain and ALPM free strain were the independent predictors of high SCD risk score.

Did you perform a follow-up to longitudinally assess the true risk of SCD/ICD shock?

• We thank the reviewer. Unfortunately long term follow-up of the patients were not done.

Was there an association of maximal LV wall thickness and papillary muscle strain?

• We thank the reviewer. Maximal LV wall thickness showed moderate correlation with anterolateral papillary muscle free strain (r=0.406, p=0.002). It has been stated in results section.

• Was an MRI available at least in some patients? Was there an association of LGE and papillary muscle strain?

• Unfortunately only small number of the patients (5 patients) had MRI studies. Since the number was so small, we could not make any statistical analysis.

In total, the incremental value of measuring papillary muscle strain, which is a sophisticated and time consuming method, does not become overt, so far.

• We thank the reviewer. According to our results ALPM free strain can be used to stratify HCMP patients for sudden cardiac risk. Risk stratification of HCMP patients could be challenging in some cases, we think that strain analysis could especially be useful in these group of subjects.

6. PLOS authors have the option to publish the peer review history of their article (what does this mean?). If published, this will include your full peer review and any attached files.

Do you want your identity to be public for this peer review? For information about this choice, including consent withdrawal, please see our Privacy Policy.

Reviewer #1: No

Reviewer #2: No

---

## [Decision Letter · Decision Letter 1]

14 Dec 2022

PONE-D-22-24350R1Does papillary muscle free strain has predictive value in risk stratification of patients with hypertrophic cardiomyopathy?PLOS ONE

Dear Dr. YILDIZ,

Thank you for submitting your manuscript to PLOS ONE. The reviewer, who is a specialist for hypertrophic CM still has major issues which must be resolved. 

We look forward to receiving your revised manuscript.

Kind regards,

Wolfgang Rudolf Bauer, M.D., Ph.D.

Academic Editor

PLOS ONE

Reviewers' comments:

Reviewer's Responses to Questions

**Comments to the Author**

1. If the authors have adequately addressed your comments raised in a previous round of review and you feel that this manuscript is now acceptable for publication, you may indicate that here to bypass the “Comments to the Author” section, enter your conflict of interest statement in the “Confidential to Editor” section, and submit your "Accept" recommendation.

Reviewer #1: (No Response)

2. Is the manuscript technically sound, and do the data support the conclusions?

Reviewer #1: Yes

3. Has the statistical analysis been performed appropriately and rigorously?

Reviewer #1: Yes

4. Have the authors made all data underlying the findings in their manuscript fully available?

Reviewer #1: Yes

5. Is the manuscript presented in an intelligible fashion and written in standard English?

Reviewer #1: Yes

6. Review Comments to the Author

Reviewer #1: The authors improved their manuscript “Does papillary muscle free strain has predictive value in risk stratification of patients with hypertrophic cardiomyopathy? But still some important points have to be mentioned:

LV outflow tract gradient measurements were only done at rest. The ESC risk stratification model includes gradients at provocations. As 50% of patients with obstructive HCM have only latent obstruction the authors underestimated risk scores in a significant number of patients without gradient measurement at provocation.

At page 5, line 21 the authors described “syncope” in general as risk factor. This is not correct as only “unexplained syncope” is an established risk factor for SCD and included in the risk stratification model. How many patients had unexplained syncope? If there are any – the statistics has to be done again.

Reference 5 describes only a weak correlation between longitudinal strain and prediction for ICD.

Page8, lines 22/23: The reported correlations are weak.

Page 9, line 4: 00.643-0.930

Page 9: Headline Discussion is missing

Page 10, lines 10-12: Reference 13 reports only weak correlations. This should be discussed.

Page 11, line 3: Reference 8 has no own data to support the statement.

Page 11, line 19: Reference 25 is no original paper, but an editorial which worked out that LGE >15% maybe an additional predictor for adverse outcome in HCM. This should be explained.

Legend of Figure 2 is wrong.

Figure 5 is not necessary

7. PLOS authors have the option to publish the peer review history of their article (what does this mean? ). If published, this will include your full peer review and any attached files.

**Do you want your identity to be public for this peer review?** For information about this choice, including consent withdrawal, please see our Privacy Policy .

Reviewer #1: No

---

## [Author Response · Author response to Decision Letter 1]

5 Jan 2023

We thank the reviewer for her/his careful reading of the manuscript and constructive remarks. We have taken the comments on board to improve and clarify the manuscript. Please find below a detailed point-by-point response to all comments.

Review Comments to the Author

Reviewer #1: The authors improved their manuscript “Does papillary muscle free strain has predictive value in risk stratification of patients with hypertrophic cardiomyopathy? But still some important points have to be mentioned:

LV outflow tract gradient measurements were only done at rest. The ESC risk stratification model includes gradients at provocations. As 50% of patients with obstructive HCM have only latent obstruction the authors underestimated risk scores in a significant number of patients without gradient measurement at provocation.

• We thank the reviewer. Measurements of LV outflow gradient were done in resting conditions. Patients who did not have resting LV outflow gradient underwent Valsalva maneuver in order to elicit latent obstruction. None of the patients in non-obstructive HCMP group had stress pressure gradients more than 30 mmHg. Patients who had labile (<30 mmHg at rest and ≥ 30 mmHg with Valsalva maneuver or other provocative tests) were not included in the study (we tried to involve patients with obstructive and non-obstructive type of HCMP). It has been explained in the material and methods section.

At page 5, line 21 the authors described “syncope” in general as risk factor. This is not correct as only “unexplained syncope” is an established risk factor for SCD and included in the risk stratification model. How many patients had unexplained syncope? If there are any – the statistics has to be done again.

• We thank the reviewer. Since only unexplained syncope is an established risk factor for sudden cardiac death in HCMP, we only included patients with a history of unexplained syncope in our study. Fifteen (27.2%) patients with HCMP had history of unexplained syncope in our study.

Reference 5 describes only a weak correlation between longitudinal strain and prediction for ICD.

• We thank the reviewer. Instead of reference 5, we used another meta-analysis which analyzed prognostic value of global longitudinal strain in hypertrophic cardiomyopathy and found that impaired left ventricular global longitudinal strain was associated with poor prognosis in HCMP patients.

Page8, lines 22/23: The reported correlations are weak.

• We thank the reviewer. In the present study we found moderate correlations of global longitudinal strain and anterolateral papillary muscle free strain with SCD risk score (r=0.480, p<0.001 and r=0.462, p<0.001, respectively). Our findings were in line with the previous data which found moderate correlation of strain values with maximal wall thickness, sudden cardiac death risk markers and other clinical and echocardiographic parameters in hypertrophic cardiomyopathy patients (1-4).

(1). Vergé MP, Cochet H, Reynaud A. Characterization of hypertrophic cardiomyopathy according to global, regional, and multi-layer longitudinal strain analysis, and prediction of sudden cardiac death. Int J Cardiovasc Imag. 2018;34(7):1091–1098.

(2). Rakesh K, Rajesh GN, Vellani H. 3D speckle tracking echocardiographic strain pattern in Hypertrophic Cardiomyopathy and its relation with Sudden Cardiac Death risk markers Indian Heart J. 2021 Jul-Aug; 73(4): 451–457. Published online 2020 Nov 20. doi: 10.1016/j.ihj.2020.11.144

(3). Abozguia K, Nallur-Shivu G, Phan TT, Ahmed I, Kalra R, Weaver RA, et al. Left ventricular strain and untwist in hypertrophic cardiomyopathy: Relation to exercise capacity Am Heart J. 2010 May; 159(5): 825–832. doi: 10.1016/j.ahj.2010.02.002

(4). Zhang L, Wan Y, He B, Wand L, Zhu D, Gao F. Left ventricular strain patterns and their relationships with cardiac biomarkers in hypertrophic cardiomyopathy patients with preserved left ventricular ejection fraction Front Cardiovasc Med . 2022 Oct 4;9:963110. doi: 10.3389/fcvm.2022.963110. eCollection 2022.

Page 9, line 4: 00.643-0.930

• We thank the reviewer. It has been corrected.

Page 9: Headline Discussion is missing

• We thank the reviewer. Headline Discussion has been added to the manuscript.

Page 10, lines 10-12: Reference 13 reports only weak correlations. This should be discussed.

• We thank the reviewer. Almaas et al. compared strain imaging with magnetic resonance imaging in detection of fibrosis in HCMP patient. They studied myectomy specimens in HCMP patients who underwent septal myectomy and histological fibrosis was classified as interstitial, replacement and total. Although they found moderate correlation of septal longitudinal strain with total and interstitial fibrosis, they did not find any correlation between late gadolinium enhancement on cardiac MRI and septal, interstitial or replacement. In that study, reduced longitudinal septal strain predicted ventricular arrhythmias in HCMP patients. According to the results of that study, compared to cardiac magnetic resonance imaging, functional assessment of myocardial functions by global longitudinal strain imaging might be a more valuable method in detecting cardiac fibrotic changes

Page 11, line 3: Reference 8 has no own data to support the statement.

• We thank the reviewer. Reference 8 has been deleted in that section.

Page 11, line 19: Reference 25 is no original paper, but an editorial which worked out that LGE >15% maybe an additional predictor for adverse outcome in HCM. This should be explained.

• We thank the reviewer. References regarding the prognostic value of strain imaging and late gadolinium enhancement in cardiac MRI studies have been added in the discussion section.

Legend of Figure 2 is wrong.

• We thank the reviewer. The order and the legends of the figures have been revised.

Figure 5 is not necessary.

• We thank the reviewer. Figure 5 has been removed.

Best regards.

---

## [Decision Letter · Decision Letter 2]

7 Feb 2023

Does papillary muscle free strain has predictive value in risk stratification of patients with hypertrophic cardiomyopathy?

PONE-D-22-24350R2

Dear Dr. YILDIZ,

We’re pleased to inform you that your manuscript has been judged scientifically suitable for publication and will be formally accepted for publication once it meets all outstanding technical requirements.

An invoice for payment will follow shortly after the formal acceptance. To ensure an efficient process, please log into Editorial Manager at http://www.editorialmanager.com/pone/ , click the 'Update My Information' link at the top of the page, and double check that your user information is up-to-date. If you have any billing related questions, please contact our Author Billing department directly at authorbilling@plos.org .

If your institution or institutions have a press office, please notify them about your upcoming paper to help maximize its impact. If they’ll be preparing press materials, please inform our press team as soon as possible -- no later than 48 hours after receiving the formal acceptance. Your manuscript will remain under strict press embargo until 2 pm Eastern Time on the date of publication. For more information, please contact onepress@plos.org .

Kind regards,

Wolfgang Rudolf Bauer, M.D., Ph.D.

Academic Editor

PLOS ONE

Additional Editor Comments (optional):

Reviewers' comments:

Reviewer's Responses to Questions

**Comments to the Author**

1. If the authors have adequately addressed your comments raised in a previous round of review and you feel that this manuscript is now acceptable for publication, you may indicate that here to bypass the “Comments to the Author” section, enter your conflict of interest statement in the “Confidential to Editor” section, and submit your "Accept" recommendation.

Reviewer #1: All comments have been addressed

2. Is the manuscript technically sound, and do the data support the conclusions?

Reviewer #1: (No Response)

3. Has the statistical analysis been performed appropriately and rigorously?

Reviewer #1: (No Response)

4. Have the authors made all data underlying the findings in their manuscript fully available?

Reviewer #1: (No Response)

5. Is the manuscript presented in an intelligible fashion and written in standard English?

Reviewer #1: (No Response)

6. Review Comments to the Author

Reviewer #1: (No Response)

7. PLOS authors have the option to publish the peer review history of their article (what does this mean? ). If published, this will include your full peer review and any attached files.

**Do you want your identity to be public for this peer review?** For information about this choice, including consent withdrawal, please see our Privacy Policy .

Reviewer #1: No

---

## [Editor Report · Acceptance letter]

14 Feb 2023

PONE-D-22-24350R2

Does papillary muscle free strain has predictive value in risk stratification of patients with hypertrophic cardiomyopathy?

Dear Dr. Yildiz:

I'm pleased to inform you that your manuscript has been deemed suitable for publication in PLOS ONE. Congratulations! Your manuscript is now with our production department.

If your institution or institutions have a press office, please let them know about your upcoming paper now to help maximize its impact. If they'll be preparing press materials, please inform our press team within the next 48 hours. Your manuscript will remain under strict press embargo until 2 pm Eastern Time on the date of publication. For more information please contact onepress@plos.org .

If we can help with anything else, please email us at plosone@plos.org .

Kind regards,

on behalf of

Prof. Wolfgang Rudolf Bauer

Academic Editor

PLOS ONE